# Using home sensing technology to assess outcome and recovery after hip and knee replacement in the UK: the HEmiSPHERE study protocol

Sabrina Grant,[1] A W Blom,[1,2] Michael R Whitehouse,[1,2] Ian Craddock,[3] Andrew Judge,[1] Emma L Tonkin,[1] Rachael Gooberman-Hill[1,2]

[1]Musculoskeletal Research Unit, Translational Health Sciences, Bristol Medical School, University of Bristol, Bristol, UK
[2]National Institute for Health Research Bristol Biomedical Research Centre, University of Bristol, Bristol, UK
[3]Department of Electrical and Electronic Engineering, Faculty of Engineering, University of Bristol, Bristol, UK

**Correspondence to**
Dr Sabrina Grant;
sabrina.grant@bristol.ac.uk

## ABSTRACT

**Introduction** Over 160 000 people with severe hip or knee pain caused by osteoarthritis undergo total hip (THR) or knee replacement (TKR) surgery each year in the UK within the National Health Service (NHS), and this number is expected to increase. Innovative approaches to evaluating surgical outcomes will be needed to respond to the increasing burden of joint replacement surgery. The Sensor Platform for Healthcare in a Residential Environment, Interdisciplinary Research Collaboration (SPHERE-IRC) have developed a system of sensors that can monitor the health-related behaviours of people living at home. The system includes sensors for the home environment (measuring temperature, humidity, room occupancy, water and electricity usage), a wristband body-worn activity monitor and silhouette (body outline) sensors. The aim of HEmiSPHERE (Hip and knEe study of a Sensor Platform of HEalthcare in a Residential Environment) is to (1) determine the accuracy and feasibility of the sensory data as it compares with conventional assessment of health outcomes after surgery using patient self-reported questionnaires, and (2) to explore how the SPHERE system is useful for everyday clinical decision-making.
**Methods and analysis** A feasibility study recruiting and installing the SPHERE system in the homes of up to 30 NHS adult patients as they undergo a THR or TKR. Through a mixed-methods design, the SPHERE system will monitor and record continuous measurements of daily behaviour. Main outcomes will assess the relationships between environmental, behavioural and movement data and the parameters of interest from the standard clinical assessments measuring patient outcomes over time. Patient interviews and focus groups with consultant orthopaedic surgeons will provide in-depth understanding of the acceptability, feasibility and accuracy of the data.
**Ethics and dissemination** We aim to disseminate the findings through regional talks and seminars, international conferences and peer-reviewed journals and social media.

## Strengths and limitations of this study

► This is the first study to install and test low-powered home sensing technology in the homes of patients undergoing joint replacement.
► Assessment of daily continuous information on patient activity in their natural setting could allow clinicians to better understand recovery, direct rehabilitation and tailor interventions.
► A mixed-methods approach to the design, evaluating sensor data against routinely used patient-reported outcome measures and qualitative information about the patient's journey strengthens the validation process.
► The present study sample is small and the patients drawn from a relatively narrow age range, which limits generalisability to patient populations in other settings.
► Although a small sample, detailed information and data analytics of each case, and exploring the perceived value of the sensor data among clinicians will enable deeper exploration about the mechanisms by which this could be integrated within current clinical systems.

Recent data from the National Joint Registry for England, Wales, Northern Ireland and the Isle of Man show the number of primary total joint replacement operations is increasing with 224 470 procedures (83 886 primary THR and 94 023 primary TKR) performed in the 12 months to 31 March 2016.[2] Trends in recent data show a median age for patients undergoing a primary TKR and THR of 70 and 69 years, respectively, and osteoarthritis as the most common reason for surgery accounting for over 90% of THRs and TKRs.[2]

With a growing, more active older population in the UK, the volume of THRs and TKRs in the UK will continue to increase.[3] Changing demographics of age, gender and Body Mass Index (BMI) and recent data indicating an

## INTRODUCTION

Total hip and knee replacements (THR and TKR) are two of the most common elective surgical procedures carried out in the UK.[1]

increasing number of younger patients undergoing THR suggest functional demands expected of these operations will change and thus assessment of outcomes may need to evolve.[4 5]

### Patient-reported outcome measures (PROMs)

PROMs are standardised, validated questionnaires completed by patients to measure functional status and well-being[6] and are now widely used as part of the national PROMs programme set by National Health Service (NHS) England.[7] PROMs are thus widely used in research and clinical settings as they are considered easy to use, inexpensive and time efficient.[8]

Assessment of health outcomes after THR and TKR focus mainly on four domains: function, mobility, pain and quality of life. Although assessment of outcomes after surgery provides evidence of effect, PROMs are subjective and, by definition, are easily influenced by socioeconomic or psychological factors and dominated by pain.[9] Furthermore, assessment measures specific to the field of orthopaedics, such as the commonly used disease-specific Western Ontario and McMaster Universities Osteoarthritis Index, validated for use with patients undergoing joint replacement, suffer (as with many orthopaedic PROMs) from a ceiling effect as it has a limited maximum value that is reached by a substantial proportion of patients who report no pain or functional limitations after surgery.[10] Bolink and colleagues highlight that a consequence of this ceiling effect is that the true extent of patient's postoperative functional availabilities cannot be determined, and further highlight the importance of investigating relative changes rather than absolute changes and to consider patients with high and low functions separately.[8] Given the limitations of PROMs, research needs to shift towards considering other methods of assessing functional outcomes after surgery.[8]

### Home Health Monitoring and 3millionlives (3ML)

In the UK, the government has been keen to promote telehealth as a way for a pressured NHS to cope with an ageing population with multiple long-term health conditions. The Whole System Demonstrator programme[11 12] was designed to demonstrate how system change alongside assistive technology could achieve a better quality of care for people living with long-term conditions and social care needs. Monitoring vital health signs remotely using telehealth was seen to enable health professionals to intervene at the right time to prevent exacerbation. Telehealth thus allows people to remain independent in their homes for longer and provides the choice and control of healthcare services to the patient. The '3millionlives' (3ML) initiative was a commitment between the Department of Health (DoH) in England and the UK telehealth and telecare industry in 2012[13] to enhance the lives of 3 million people over the next 5 years by accelerating the roll-out of telehealth and telecare in the NHS and social care. In turn, 3ML would

reduce the burden of acute hospital inpatient care and deliver more cost-effective services as part of a modern model of integrated care. The programme rolled out through a wide range of delivery modes and mechanisms from creative innovation, new housing solutions, integration of services, interoperable systems, smart cities and mobile technologies (digital patient records and data protection).

### Technology Enabled Care Services (TECS) (NHS England) Evidence database

The TECS programme[14] was born out of the 3ML initiative. The programme has resulted in mixed findings[15]; however, a positive outcome was the development of a large and complex evidence database. TECS are studies of complex interventions involving people, processes and technology, based on a range of methodologies; therefore, results are influenced by each of those elements. In some contexts, there are mixed messages on the clinical and cost-effectiveness of some of these TECS.

Three key studies from the database highlight the empirical evidence around the use of smart wearable body sensors for patient self-assessment and monitoring,[16] the impact of digital engagement on the quality of life of older people[17] and the technical challenges of developing high-quality biometric signals from unsupervised patients at home.[18] Studies investigating wearable body sensors such as connected devices, trackers, telemonitoring, wireless technology and real-time home tracking devices demonstrate that smart wearable sensors are effective and reliable for preventative interventions in many different facets of medicine such as cardiopulmonary, vascular, endocrine, neurological function and rehabilitation medicine.[16] Such sensors have also been shown to be accurate and useful for perioperative and rehabilitation medicine.

Many individuals with chronic disease could benefit from having constant remote monitoring, and the best way to monitor a patient is through understanding their interactions with their environment and daily activities. Giving the patient the opportunity to leave the hospital or healthcare environment but continuing to monitor them allows for a more authentic representation and a more accurate assessment of physiological data. For patients who undergo THR and TKR, the ability to reliably monitor away from the hospital could decrease the healthcare and cost burden associated with inpatient length of stay and outpatient follow-up.

There is consensus within the literature that incorporating smart wearable sensors into routine care of patients could augment physician–patient relationships, increase the autonomy and involvement of patients with their healthcare and provide for novel remote monitoring techniques, which could revolutionise healthcare and management. The future challenges lie within integrating telehealth within older populations[17] and applying predictive analytics and data mining for enhancing

clinical decision support,[18] which HEmiSPHERE, in part, aims to address.

## Sensor Platform of Healthcare in a Residential Environment— SPHERE

The Sensor Platform of Healthcare in a Residential Environment Interdisciplinary Research Collaboration (SPHERE-IRC) at the University of Bristol, Faculty of Engineering, have developed innovative technology comprising a group of low-power sensors that can measure information continuously about the home (eg, temperature, energy consumption etc) as well as information about people in the home (eg, location, how active they are, extent of movement) and their health-related behaviours. A key element to the SPHERE sensor system is that data are time-stamped and collected within their natural home environment.

It is proposed that this 'real time' continuous data collected by SPHERE sensors could potentially, in contrast to conventional methods of assessment of patient change and recovery processes through the collection of PROMs, capture a more global picture of the patient as they undergo a THR or TKR. The SPHERE 100 Homes study has been designed and is being conducted by SPHERE-IRC to try out and test this system in different populations, generate data and explore whether the system works.

## Hip and KnEe Study of a Sensor Platform of Healthcare in a Residential Environment (HEmiSPHERE)

The present HEmiSPHERE study is embedded within the SPHERE 100 Homes study. The aim of HEmiSPHERE is to understand whether clinically useful information can be obtained from the technology enabled service in a cohort of orthopaedic patients undergoing elective surgery (THR or TKR). Understanding how these data can help shape future healthcare from the perspectives of the end users (the patient and the clinician) and determining whether this form of telehealth enables or constrains clinical decision-making also remains the primary purpose of this research study.

Specifically, this project has four main objectives:
a. To determine if the raw sensor data obtained from the SPHERE system can be processed to translate into meaningful data for the clinician.
b. To understand how the information provided by the SPHERE system enables, shapes or constrains clinical decision-making from the health professionals' perspective and explore whether the SPHERE system is feasible and acceptable through the perspectives of patients' undergoing elective surgery (THR or TKR).
c. To assess whether data from the SPHERE system provide a comparable measure of patient function and mobility than routinely collected standardised patient measures of PROMs.
d. To determine factors that facilitate or hinder the installation of the SPHERE system.

# METHODS AND ANALYSES
## Study design

This feasibility study will adopt a concurrent mixed methodology approach to understand how clinicians could use the SPHERE system in their everyday clinical decision-making.[19] The quantitative approach will consist of comparing and contrasting data collected from standardised PROMs to that collected from the SPHERE sensors installed within patients' homes. In parallel, the study will qualitatively explore patients' experiences with the technology through in-depth interviews and explore the clinical value of the data through focus groups with health professionals. Bringing together information from both these sources will offer a more detailed understanding of whether the information collected by this novel technology could potentially improve healthcare delivery (see figure 1).

## Study setting

Installation of the SPHERE sensors will take place in the homes of NHS orthopaedic patients from one region of Bristol, UK. Interviews will be held within the patients' homes. Focus groups with health professionals will be held within the Trust they are employed within.

## Participants

Patient participants.

### Inclusion criteria
► Patients aged 18 years and over.
► Patients referred for a THR or TKR.
► Patients able to read and understand English.

### Exclusion criteria
► Lack of capacity to consent.
► Patients where a child (under 16 years) resides in their home.

## Household participants

The study involves installation of the SPHERE system (sensors) within the patient's house; therefore, all participants residing within the household (family members or other key significant others) will become 'participants' in order that the SPHERE sensors are able to identify and differentiate the patient from the other key significant others. These participants will be referred to as household members and will each individually provide consent.

## Health professional participants

We will recruit a convenience sample of orthopaedic surgeons employed within one NHS trust in Bristol, who perform joint replacement operations to participate in the focus groups. The focus group will be held approximately 6 to 9 months after the first patient is recruited.

## Sample size

This is an exploratory study and therefore numbers of participants are chosen on pragmatic grounds and based on the experiences of the pilot installations from

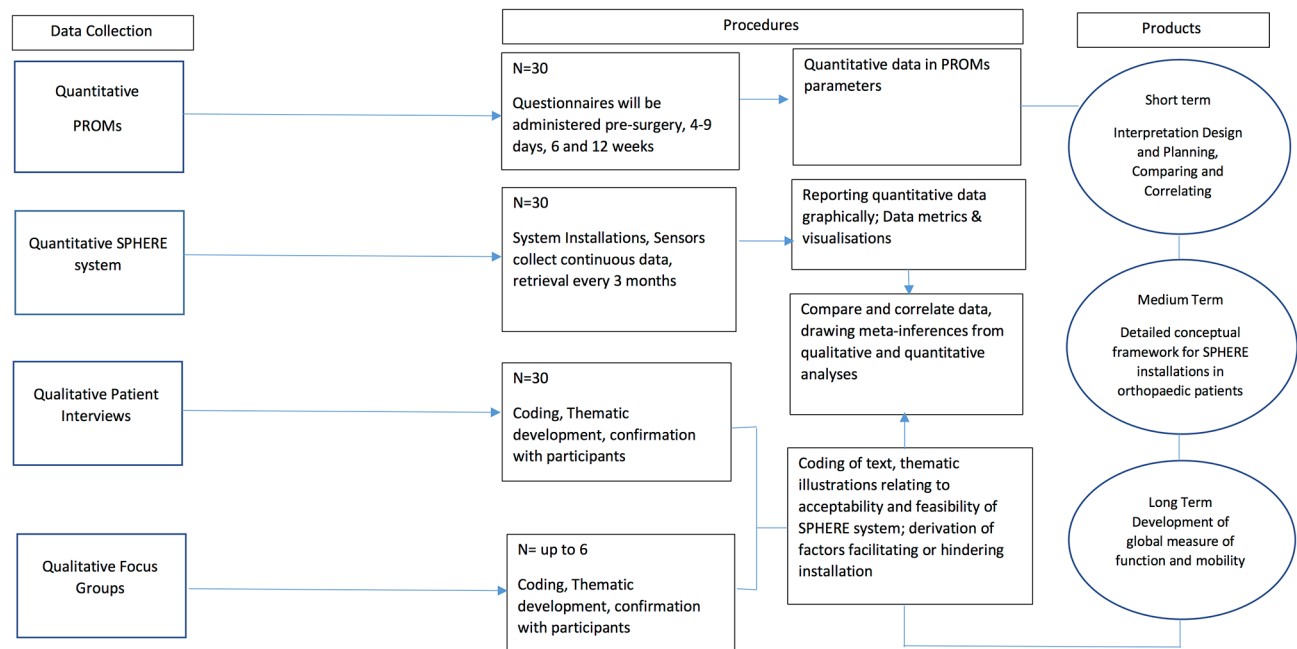

**Figure 1** Recruitment and consent process. PROM, patient-reported outcome measure; SPHERE, Sensor Platform of Healthcare in a Residential Environment.

the SPHERE 100 Homes study. We will include as many eligible participants as possible. For individual outcome data collection, we aim to allow for up to 30 participants undergoing a THR/TKR to be recruited.

Up to 30 patients will have the SPHERE system installed within their home and will each participate in the three forms of data collection, that is, questionnaires, interviews and collection of SPHERE sensor data, as depicted in figure 2. Each participant will complete two interviews and therefore up to 60 interviews may be conducted. Patients may be identified during the study as unsuitable for varying reasons (eg, unsuitable house, health complications affecting waiting time for surgery). Based on previous studies, we would expect to approach up to 100 patients to achieve a sample size of 30.

## Data collection

### Qualitative strand

#### Interviews

Pre-surgery, the purpose of the in-depth interview is to provide a baseline understanding about the patients' general attitude to health technology and current usage of existing health technology in their home, for example, use of home blood pressure monitor and use of health mobile apps. This is a conventional exploratory qualitative analysis. Post-surgery, qualitative analysis will take a more directed approach so that findings can be triangulated with information provided by the PROMs and from the sensors. The purpose of triangulation is not necessarily to cross-validate data but to capture different dimensions of the same phenomenon.[20] Strength of qualitative research is the benefit of induction, where the unexpected is discovered through interpreting the data.[21] This is a process whereby you move forward

the claims about the observed (which we will get from the SPHERE sensors and the summative data from the PROMs) to the unobserved. Qualitative research is concerned with generating 'credible' and 'trustworthy' data and analysis, involves interpretation and induction/abduction, the key benefits of combining this source with that of quantitative PROMs and continuous sensory data provide a stronger platform for demonstrating the robustness of our interpretation of the data, and that our reasoning is credible.

Questions will incorporate open-ended techniques (eg, 'can you tell me about' or 'talk me through') and the use of probes (eg, can you tell me more about that?). During the interview, the researcher will use a topic guide developed in collaboration with the research programme's Public and Patient Involvement group and based on existing literature and theoretical underpinnings for living with home monitoring technology. The topic guide will follow the structure outlined in boxes 1 and 2. Topics will include current use of health technology within the home, expectations of surgery in relation to mobility and function, and experiences of living with the SPHERE system. The topic guide will be refined as data collection progresses and will be piloted in the first three to four interviews.

### Focus groups

The purpose of the focus group is to explore, using a topic guide (see box 3), health professionals' views and experiences of the two types of data provided by the PROMs and the SPHERE sensors. HEmiSPHERE is a study that involves iterative processes, where health professionals will compare both forms of data, which will help to inform and refine further how the SPHERE system data can be

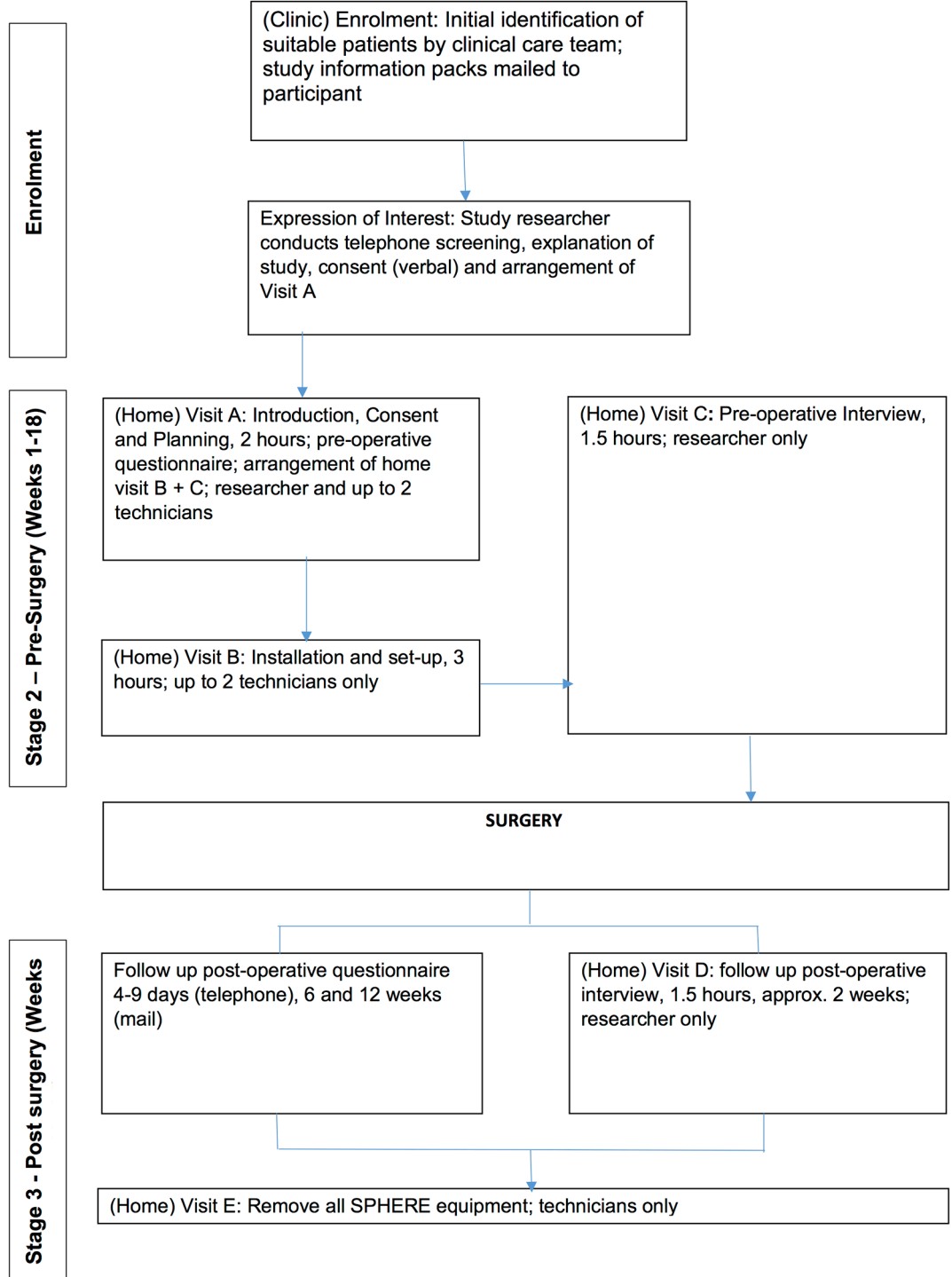

**Figure 2** Study flow chart for patients. SPHERE, Sensor Platform of Healthcare in a Residential Environment.

| Box 1   Topic guide—preoperative (patient) |
|---|
| ► Case history to referral for surgery. |
| ► Household constitution. |
| ► Experience of health technology (home, wearable, apps). |
| ► Current experience and future expectations of mobility and function. |
| ► Preparations in the household for surgery. |

| Box 2   Topic guide—postoperative (patient) |
|---|
| ► Experience of living with SPHERE technology. |
| ► Relate data from individual PROM assessments within the questions. |
| ► Ask about the adequacy of information received about SPHERE technology. |
| ► Explore how initial expectations of living with the SPHERE technology compares with those after surgery. |

 

**Box 3  Topic guide—postoperative (health professional)**

- ► Stimulus-based discussion of the two data sources (PROMs and output of the SPHERE sensors).
- ► Transforming the data.
- ► Relevance of each sources to clinical decision-making.
- ► Compare the two data sources on accuracy, reliability and usefulness.

transformed and provide anything useful beyond more conventional assessments of PROMs.

### Quantitative strand
#### Patient-reported outcome measures
*Baseline and follow-up data*

The baseline questionnaire will include a collection of basic demographic information (marital status, occupation, ethnicity, education and age) and a series of self-reported outcome measures to assess general health, quality of life, pain and functional mobility specific to the hip or knee and psychological status. The Hospital Anxiety and Depression Scale[22 23] is a 14-item self-rating scale commonly used by doctors (and other health professionals) to determine the levels of anxiety and depression that a patient is experiencing. The EuroQol scale[24 25] (EQ-5D-5L) is a quality-of-life instrument that comprises five dimensions (mobility, self-care, usual activities, pain/discomfort and anxiety/depression) and a visual analogue scale for respondents to self-rate their health state. The Short Form 12[26 27] (SF-12) is a generic measure to provide measurement of mental and physical functioning and overall health-related quality of life. The combination of the EQ-5D-5L and the SF-12 provides relatively broad coverage of important health domains and scores for various purposes. The Pittsburgh Sleep Quality Index is a 19-item measure to assess sleep quality.[28]

Hip or knee pain and functional mobility will be measured using the Oxford Hip Score[29] or Oxford Knee Score[30]. These measures are joint-specific 12-item questionnaires asking patients a series of questions that allow them to rate their level of pain and aspects for functional ability following surgery. The Knee Injury and Osteoarthritis Outcome Score (KOOS) and the Hip disability and Osteoarthritis Outcome Scores[31 32] (HOOS) assess pain, other symptoms, function in daily living, function in sport and recreation, and hip-related or knee-related quality of life. The HOOS and KOOS are used to assess the patient's opinion about their hip or knee and associated problems. We will be using the quality-of-life subscale within these measures.

The procedure and assessment times to measure these outcomes at baseline and subsequent time points are shown in table 1. Follow-up calls will start 4–9 days after discharge from hospital as outlined in table 1. In the event

**Table 1**  Baseline and follow-up data collection time points

| Time point | Outcomes measured | Tools |
|---|---|---|
| Clinical identification | Baseline demographic information | Clinical inclusion eligibility criteria |
| First telephone eligibility screening—preoperative | Confirm baseline demographic information, home fabric and eligibility | Telephone screening eligibility questionnaire |
| Visit A—preoperative, self-completion | HrQoL, function status, pain, anxiety and depression, patient self-rating of hip or knee function | EQ-5D-5L, SF 12, HADS, OHS or OKS, HOOS/KOOS (QoL subscale) |
| Visit C—preoperative interview | Questions relating to history to referral for surgery, household constitution, experience of home health technology, household preparation and constitution, expectations of mobility and function | Topic guide box 1 |
| First telephone follow-up (4–9 days after hospital discharge) | Baseline demographic information, HrQoL, function status, pain, anxiety and depression, patient self-rating of hip or knee function, sleep routines/habits | EQ-5D-5L, SF-12, HADS, OHS or OKS, HOOS/KOOS (QoL subscale), PSQI |
| Second telephone follow-up (6/52) | Baseline demographic information, HrQoL, function status, pain, anxiety and depression patient self-rating of hip or knee function, sleep routines/habits | EQ-5D-5L, SF-12, HADS, OHS or OKS, HOOS/KOOS (QoL subscale), PSQI |
| Third telephone follow-up (12/52) | Baseline demographic information, HrQoL, SF-12, pain, HADS, patient self-rating of hip or knee function HADS, patient self-rating of hip or knee function, sleep routines/habits | EQ-5D-5L, SF-12, HADS, OHS or OKS, HOOS/KOOS (QoL subscale), PSQI |
| Visit D—2-week postoperative interview | Questions relating to experience of living with SPHERE technology relating to PROMs | Topic guide box 2 |
| 6–9 month postoperative health professional focus group | Questions relating to the usability of SPHERE technology data compared with PROMs | Topic guide box 3 |

EQ-5D-5L, EuroQol 5D-5L; HADS, Hospital Anxiety and Depression Scale; HOOS/KOOS, Hip/Knee Dysfunction Osteoarthritis Outcome Score; HrQoL, health-related quality of life; OHS, Oxford Hip Score; OKS, Oxford Knee Score; PROM, patient-reported outcome measure; PSQI, Pittsburgh Sleep Quality Index; SF-12, Short Form 12; SPHERE, Sensor Platform of Healthcare in a Residential Environment.

of delays in patients being discharged from hospital, the study will adapt and build in flexibility on an individual basis.

### SPHERE sensor system

The University of Bristol's SPHERE project will provide equipment and analytical means to determine the person's physiological envelope classified by activity types and levels. The study involves participants living with the SPHERE sensor system comprising sensors for the home environment (measuring temperature, humidity, room occupancy, water, electricity usage), a wristband body-worn activity monitor and silhouette sensors.

### Sensor data

This data collection happens continuously unless the participant uses the SPHERE Genie (described below) to turn off the system.

### Environmental sensors

These small boxes (about the size of a fire alarm) include room presence sensors (as used by security alarms), humidity, temperature, water and flow metres. Small sensors will be plugged to some appliances (kettle, microwave and fridge) to measure electricity use.

### Wearable devices

The patient participant will be asked to wear a wristband; a research-quality accelerometer designed by the University of Bristol SPHERE team, tuned in to both hardware (physical electronics) and software (the programs and other operating information used by a computer). Inside the house, these sensors help the SPHERE system to measure movement, including the room the participant is in. Outside the home, they do not track the participant but only whether the participant is active or not.

### Motion-analysis sensors

Two or three areas of the house will be equipped with a small motion-analysis sensor and an associated small computer (the size of a paperback book). There will be no motion-analysis sensors in the bathroom or bedrooms. The motion-analysis sensors do not record video; they capture features such as the position, orientation and silhouette (body outline) of any person in view of the sensor.

### SPHERE Genie

The SPHERE Genie is the participant control panel that allows the participant to control the SPHERE system. Participants can delete data up to 24 hours earlier or pause the system and allows the participant to contact the SPHERE team. Frequency of deletion of data will be logged for monitoring purposes in order to address the magnitude of deletion of data. The SPHERE team will respond to the queries within a working day. The SPHERE team and the participant will communicate through the Genie using the participants email address or mobile telephone number.

### Activity data

Participants will be requested to allow information to be collected about their activities of daily living detected by the wrist-worn and environment sensors. These data will be captured in the form of silhouette motion analysis tuned into the requirements of orthopaedic clinicians where possible, at the same time as the periodic assessment of PROMs. These data will be collected for the entire duration of study participation as outlined in figure 1. The main outcome of the study is to compare this continuous activity data with data that are routinely collected from standardised PROMs when a patient undergoes a THR or TKR. This process will be iterative in that clinicians will be presented with data visualisations that the SPHERE sensor system currently produces, they will feedback with developing concepts and refine requirements. Therefore, the relationship investigated will be between the patients' level of activity and function and how this changes over time and the various factors important for optimum recovery. How the SPHERE sensory system improves understanding of that relationship will be investigated.

### Activity monitoring

The wearable wrist-worn device is designed to be minimally intrusive. The wearable device measures three-axis acceleration to an accuracy of 32 milli-g (approximately $0.3 \, \text{m/s}^2$). The indoor position measurement with the wearable device is to an accuracy of around 5 m (depending on room layout). The size of the sensor is no larger than an average watch.

The devices communicate encrypted data using commercial low-power wireless protocols. Each patient's house will be equipped with two or three wireless access points for use with the SPHERE box, powered by standard mains-driven domestic USB power supplies. These have only transient physical contact with patients (for device re-charging); this hardware is off the shelf and CE-marked for domestic use.

The SPHERE devices, apart from communicating over Bluetooth, also communicates over the University of Bristol's own Wi-Fi network (we provide a separate router that operates at 5 GHz frequency, instead of 2.4 GHz, which is most common for home networks). The SPHERE team will not use the participant's home broadband; instead, we use LTE routers to send diagnostic information over the 3G/4G network. This is only for maintenance tasks such as checking that the sensors are running and battery levels. Dynamic channel changing will be used to avoid interference with home networks.

### Outcomes

Quantitative outcomes will be assessed with data metrics of the SPHERE system (measured continuously) and with the PROMs questionnaires (completed at specific time points before and after surgery). Individual quantitative changes will be interpreted by comparison with qualitative interview and focus group data. These will be in the form of reported associations between these two types of

data before surgery (preoperative), and 4–9 days, 6 and 12 weeks after surgery (postoperative). Other parameters of interest will include quality of life, hip/knee pain, hip/knee symptoms, psychological status and quality of sleep assessed by validated standardised measures.

The development of a comparable measure of patient function and mobility will be informed by the quantitative and qualitative strands of the project. To understand whether living with the system is feasible and acceptable for patients undergoing surgery, and to address the factors that facilitate or hinder installation will be informed by the qualitative interviews.

## Recruitment process

### Identification of patients

The clinical care team will identify potential participants who are eligible for the study. Patients will be approached consecutively and mailed a study information pack (letter of invitation, an information booklet, a reply slip and a FREEPOST addressed envelope). The information booklet will describe the purpose and aims of the study stating that they should return the reply slip indicating whether they are interested in taking part. If no response is received within 7 days, the clinical care team will send out a single reminder and information pack (supporting document 2). To track those that do not reply to the reminder and to provide statistics on the non-responders, the clinical care team will record anonymised data on age and gender for all patients sent a study information pack.

### Enrolment

After an expression of interest is received, participants will be contacted by telephone and verbal agreement to participate confirmed and an appointment will be arranged to conduct Visit A (Introduction, Consent and Planning) by the study research team (study researcher and up to two SPHERE technicians) (see figure 2).

Participants will be informed that Visit B will be to install the SPHERE technology, the date and time of which will be arranged with the participant at Visit A.

### Interviews

Interviews will be held at Visits C and D (see figure 2). Participants will be contacted by telephone and invited to participate in an interview. If the participant agrees to be interviewed, a suitable time and date will be arranged and a confirmation letter will be sent to the participant.

### Household members

Consent will be taken from each household member during Visit A using a separate participant information sheet and consent form.

The recruitment and consent process are depicted in figure 3.

### Recruitment period

Recruitment will run for approximately 8 months with an anticipated start date of September 2017.

## Data analysis

Data collection will take place throughout the 12-month study duration (see table 1). Data from the PROMs questionnaires will be entered onto a secure database. The SPHERE system will provide continuous data on the normal activities and movements of the participants within their home.

The study researcher and study statistician will work collaboratively with the clinicians and the SPHERE engineering team to determine how the continuous SPHERE data collected from wearable wrist monitors, environment sensors and motion analysis (silhouette movements) can be transformed into various summary variables (eg, peak activity level, disruption of sleep time points). This will allow us to ascertain the extent to which these data correlate with the data provided by PROMs at relevant time points. Descriptive summary statistics (mean (SD), number (%), correlation) and statistical techniques (regression modelling) will be used to compare the data sets from the PROMs and SPHERE sensor system and report any associations. Determining predictive analytics for enhancing clinical decision support will subsequently form the latter phase of data analysis.

Interviews will be audio-recorded on a digital encrypted recording device, transcribed, anonymised and imported into qualitative software package QSR NVivo, v.11. Data will be analysed thematically, involving inductive coding and categorisation.[33] A sample of approximately 20% of the transcripts will be double-coded by another member of the research team.

## Study limitations

The present study sample is small and the patients drawn from a relatively narrow age range, which limits generalisability to patient populations in other settings. The nature of the research is exploratory. Deploying this technology as patients undergo surgery has implications on participant burden and may challenge recruitment. A key focus of the present study will be to address and consider such challenges, barriers and levers in conducting this research study within NHS ethical and governance frameworks. The feasibility nature of the present study means that cost-effectiveness of the system will not presently be explored, but forms part of our wider programme of future research.

## Data integration

Owing to the mixed methodology design, the qualitative and quantitative components will be analysed separately but concurrently. This is the method of choice when analysis of one strand is necessary to inform the conduct of the next.[19] Inferences will be drawn from both strands and across strands. These meta-inferences will be used to draw conclusions. Figure 1 outlines the steps for data analysis and integration.

## Data storage and sharing

The day-to-day management of the study and study data will be co-ordinated by the study researcher in the

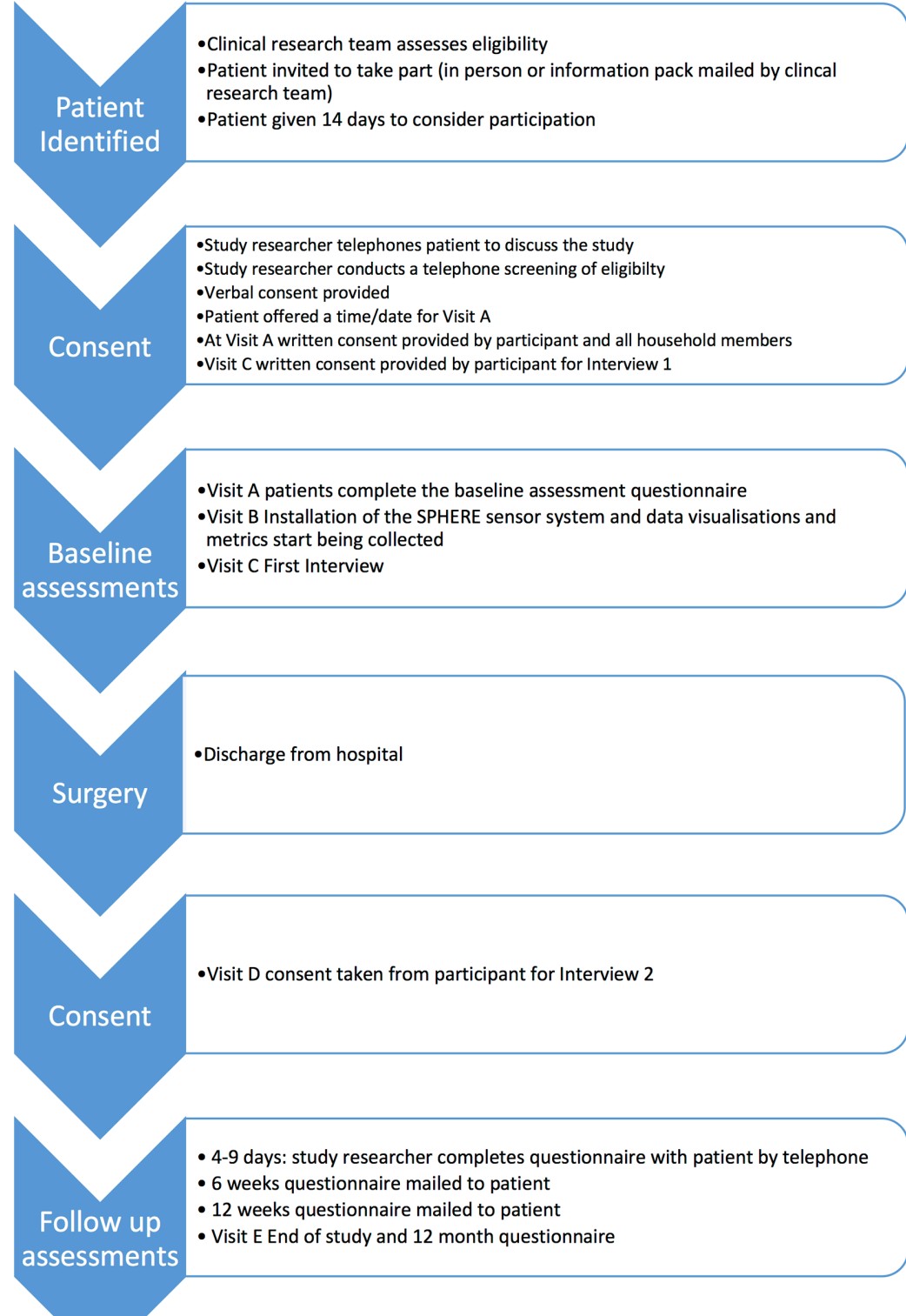

**Figure 3** Overview of analysis and products of different strands in a concurrent design. SPHERE, Sensor Platform of Healthcare in a Residential Environment.

Musculoskeletal Research Unit. Periodically, data will undergo additional checks to ensure consistency between data submitted on Case Report Forms (CRFs). The study sponsor will monitor the study, and this will include reviewing the Site File and patient medical records.

Anonymised data will be securely stored within the University of Bristol and shared according to University of Bristol procedures and guidelines.

Data procedures will be in keeping with the stipulations in the General Data Protection Regulation. In line

with NIHR guidance, which encourages the sharing of anonymised datasets (for further information, please see http://www.journalslibrary.nihr.ac.uk/replace/report-preparation/publication-ethics/3), we will be seeking consent from patients for their data to be shared anonymously with other researchers.

## Confidentiality

All participants will be assured of the confidentiality of the data collected, but will be asked during the consent process for their permission to publish anonymised quotations from the study. Participant paperwork, including consent forms and reply slips, will be maintained in a locked filing cabinet in the University of Bristol's Musculoskeletal Research Unit, which is a secure unit with card-controlled access.

## Patient and public involvement

### Patient Experience Partnership in Research (PEP-R)

This study was developed in collaboration with the University of Bristol's Musculoskeletal Research Unit's patient involvement groups, Patient Experience Partnership in Research (PEP-R).[34] PEP-R comprises nine patients with musculoskeletal conditions, most of whom have had joint replacements, all of whom have had experiences of long-term pain, some after knee replacement.

The group is supported by the Research Unit's experienced Patient and Public Involvement co-ordinator (AB). We also consult regularly with SPHERE's Public Engagement Associate to discuss the design of materials and any other relevant documentation (eg, patient information leaflet, use of SPHERE sensor system images and interview topic guide). Towards the end of the study, both groups will also be consulted to discuss anonymised findings and dissemination strategies.

## Publication policy

All collaborators will take an active part in the preparing and reviewing of all manuscripts and reports generated during or because of this study. All publications will accord with revised Open Access policy in the Research Councils UK guidance (http://www.rcuk.ac.uk/research/openaccess/).

## Dissemination policy

In addition to provision of annual and final reports, as well as presentations at scientific meetings and publication of findings in scientific literature, all participants in the study who want it will be sent a summary of the final results of the study. A copy of any related journal articles will also be available on request from the sponsor. Depending on the final findings of the study, the investigators/collaborators may also seek further dissemination to both potential service users and professionals.

**Contributors** SG, MRW, AWB, IC and RG-H contributed to the conception of the study. SG developed the first draft of the manuscript. SG, AJ and ELT provided statistical and methodological input. SG, MRW, AWB, AJ, IC, ELT and RG-H read and edited several versions of the manuscript, provided critical evaluation and have given final approval of the manuscript.

**Funding** The study is funded by the Engineering Physical Sciences Research Council (EPSRC) (grant no. EP/K031910/1).

**Competing interests** None declared.

**Patient consent** Not required.

**Ethics approval** South West—Central Bristol Research Ethics Committee (17/SW/0121) on 22 June 2017.

**Provenance and peer review** Not commissioned; externally peer reviewed.

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
