## [Reviewer comments · BMJ Open]

ARTICLE DETAILS

TITLE (PROVISIONAL)	Using home sensing technology to assess outcome and recovery after hip and knee replacement in the UK: the HEmiSPHERE study protocol
AUTHORS	Grant, Sabrina; Blom, AW; Whitehouse, Michael; Craddock, Ian; Judge, Andrew; Tonkin, Emma; Gooberman-Hill, Rachael

VERSION 1 – REVIEW

REVIEWER	Ilana Ackerman Monash University, Australia
REVIEW RETURNED	08-Feb-2018

GENERAL COMMENTS	Review of Grant et al: Hip and KnEe study of a Sensor Platform of Healthcare in a Residential Environment (HEmiSPHERE) – Study Protocol This protocol paper describes an interesting and novel study that will use a home-installed sensor system to monitor patient outcomes after hip and knee replacement surgery. In particular, the proposed technology to support home-based monitoring is very impressive and the mixed-methods approach will be useful for evaluating feasibility as well as patient and clinician perspectives. While the protocol is very comprehensive, there are several areas that require further clarification and consideration. Abstract: - it would be helpful to include a sentence (after describing the increasing burden of joint replacement surgery in the UK) stating that innovative approaches to evaluating surgical outcomes will be needed given the growing burden. This would then lead on well to the description of SPHERE and the HEmiSPHERE study- I would describe this study as a pilot or mixed-methods feasibility study rather than an observational cohort study (given the mixed-methods approach and small sample size) Strengths and limitation - the third point could also refer to the perceived value of the sensor data among clinicians
--

Introduction

- Page 4, line 12: assessment of health outcomes after joint replacement really focuses on 4 key domains: function, mobility, pain, and quality of life
- Page 4, line 54: roll out of telehealth and telecare could increase 'efficiency' rather than cost-effectiveness (the costs associated with developing and rolling out these new systems will be substantial so unlikely to translate into improved cost-effectiveness considering already very good outcomes from joint replacement surgery)
- Page 5, line 3: it's not clear what 'creative innovation new housing solutions', 'interoperable systems' or 'smart cities' means
- Page 5, line 49: not sure how incorporating smart sensors into routine care would 'augment physician-patient relationships' given patients would presumably be spending less time with their physician?

Methods

- Page 7, line 8: Again I would specify up front that this is a pilot feasibility study or a mixed-methods feasibility study
- Page 8, line 6: The authors may want to consider including other key clinicians (eg physiotherapists) in the focus group sample, given that they have more frequent contact with patients after joint replacement surgery with regard to their rehabilitation (as opposed to the orthopaedic surgeons who most commonly review the patient at one specified post-operative appointment)
- Page 8, line 49: can the authors clarify what is meant by 'comparing for similarities and differences'?
- Page 10, line 15: will the surgeons have seen the PROMs and sensor data prior to the focus group to enable them to compare the data sources (as per Box 3)?
- Page 10, line 38: the authors could leave out the SF-12 measure given they are already collecting EQ-5D data and mindful of respondent burden as participants will also be completing the disease-specific measures
- Page 12, line 12: it's not clear why participants will have the capability of deleting data from the SPHERE system: how will this be tracked and how might it affect sensor data completeness?
- Page 14, Table 1 – the sleep quality questionnaire is not mentioned in this table
- Page 14, Table 1 – am querying whether the First telephone follow-up at 4-9 days post-hospital discharge is needed? This is a very early post-operative time point when most patients will still have significant pain and may not want to undergo the telephone survey (the possibility of missing data should be considered). I think it would be reasonable to make the 6 week telephone call the first post-operative follow-up
- Page 15, line 16: it's not clear why a global measure of patient function and mobility will be developed and how this will be used (will this be used to compare to sensor data?)
- Page 16: while the disruption of sleep data can be compared directly to the sleep quality questionnaire data, I'm concerned that the activity data collected from the sensors is really measuring something different to that collected with the other PROMs tools (I note that no standardised physical activity tool is

	being used). Activity levels are a very different construct to hip- and knee-related function (collected by the OHS/OKS, with regard to specific functional activities including getting in/out of a car or public transport, doing the household shopping, pain in bed at night). The EQ-5D only includes 1 mobility item and the HOOS/KOOS quality of life subscales don't refer specifically to mobility or physical activity at all. Given this, I'm not sure that it makes for a fair comparison and would like the authors to clarify exactly how they will be comparing the sensor data and provide justification for why this is a reasonable approach Ethics and dissemination  - Page 18, line 20: this should read 'a single site study' Other points:  - A brief section summarising the study limitations should also be included - It would be also valuable to report costs data if available (eg average cost per patient, with regard to the costs of equipment, installation and ongoing monitoring and IT support)
--	---

REVIEWER	Ola Rolfson Department of Orthopaedics, Institute of Clinical Sciences, Sahlgrenska Academy, University of Gothenburg
REVIEW RETURNED	07-Mar-2018

GENERAL COMMENTS	This study protocol describes a pilot study to explore the use of a home sensor system to measure activity and motion in patients eligible for hip or knee replacement, before and after surgery. It will investigate how data from the home sensor can be used as alternative/complement to patient-reported outcome measures. It will also investigate the feasibility and acceptability of using home sensors with a qualitative approach. This is innovative research. Current methods to measure the primary outcomes of joint replacement have limitations. The proposed home sensors may complement patient-reported outcome measures and provide a more nuanced understanding of how patients improve their activity following joint replacement.
---

VERSION 1 – AUTHOR RESPONSE

Reviewer 1:

Thank you for your comments, we have now changed the figures to a better resolution

Reviewer 2:

This protocol paper describes an interesting and novel study that will use a home-installed sensor system to monitor patient outcomes after hip and knee replacement surgery. In particular, the

proposed technology to support homebased monitoring is very impressive and the mixed-methods approach will be useful for evaluating feasibility as well as patient and clinician perspectives. While the protocol is very comprehensive, there are several areas that require further clarification and consideration.

Abstract:

- it would be helpful to include a sentence (after describing the increasing burden of joint replacement surgery in the UK) stating that innovative approaches to evaluating surgical outcomes will be needed given the growing burden. This would then lead on well to the description of SPHERE and the HEmiSPHERE study

Thank you for your comment, we have added a sentence to the abstract.

- I would describe this study as a pilot or mixed-methods feasibility study rather than an observational cohort study (given the mixed-methods approach and small sample size)

Though this is an observation of a cohort of patients over a period of months, we agree with the reviewer that this is a feasibility study using a mixed methods approach. We have therefore changed the wording on the abstract to say 'A feasibility study....

Strengths and limitation

- the third point could also refer to the perceived value of the sensor data among clinicians

Thank you for your recommendation, instead of including this in point 3, we have amended the points to now read:

- The present study sample is small and the patients drawn from a relatively narrow age range which limits generalisability to patient populations in other settings.
- Although a small sample, detailed information and data analytics of each case, and exploring the perceived value of the sensor data among clinicians will enable deeper exploration about the mechanisms by which this could be integrated within current clinical systems.

Introduction

- Page 4, line 12: assessment of health outcomes after joint replacement really focuses on 4 key domains: function, mobility, pain, and quality of life

Thank you for your comment and have now added quality of life. Sentence now reads:

“Assessment of health outcomes after THR and TKR focus mainly on four domains: function, mobility, pain and quality of life.”

- Page 4, line 54: roll out of telehealth and telecare could increase ‘efficiency’ rather than cost-effectiveness (the costs associated with developing and rolling out these new systems will be substantial so unlikely to translate into improved cost-effectiveness considering already very good outcomes from joint replacement surgery)

Thank you for your comment. This was a statement or mission of the ‘3millionlives’ initiative. We have altered the sentence to improve clarity of the sentence. It now reads:

The ‘3millionlives’ (3ML) initiative was a commitment between the Department of Health (DOH) in England and the UK telehealth and telecare industry in 2012, [13] to enhance the lives of three million people over the next 5 years by accelerating the roll-out of telehealth and telecare in the NHS and social care. In turn 3ML would reduce the burden of acute hospital inpatient care and deliver more cost-effective services as part of a modern model of integrated care.

- Page 5, line 3: it’s not clear what ‘creative innovation new housing solutions’, ‘interoperable systems’ or ‘smart cities’ means

Thank you for your comment. These refer to the number of wide ranging interventions that were launched from the ‘3millionlives’ programme. The following section about the Technology Enables Care Services (TECS) encompasses some examples, and so we have taken out this sentence.

- Page 5, line 49: not sure how incorporating smart sensors into routine care would ‘augment physician-patient relationships’ given patients would presumably be spending less time with their physician?

Smart wearable sensors are effective and reliable for preventative methods in many different facets of medicine such as cardiopulmonary, vascular, endocrine, neurological function and rehabilitation medicine. Such interventions are listed within Appelboom et al’s (2014) review – reference 16 within the manuscript. Though it is correct that smart sensors may mean less time spent face to face with the clinician, sensors that provide a closed-loop system for patients such as glucose home-monitoring in the management of diabetic patients, or home monitoring of blood pressure in patients with hypertension, lends itself to more personalized medicine to patient’s in novel ways that were not available before. Alternative to strict monitoring, these devices have the ability to calculate idiosyncratic patterns that can be used to modulate treatment and tailor it to specific needs of the individual.

Methods

- Page 7, line 8: Again I would specify up front that this is a pilot feasibility study or a mixed-methods feasibility study

Thank you, we have amended the sentence to:

This feasibility study will adopt a concurrent mixed methodology approach to understand how clinicians could use the SPHERE system in their everyday clinical decision making.[19]

- Page 8, line 6: The authors may want to consider including other key clinicians (eg physiotherapists) in the focus group sample, given that they have more frequent contact with patients after joint replacement surgery with regard to their rehabilitation (as opposed to the orthopaedic surgeons who most commonly review the patient at one specified post-operative appointment)

We think this is a great suggestion from the reviewer, and plan to undertake focus groups with other allied health professionals in our future work. For the present study, timelines permit inviting orthopaedic surgeons only to the present study as we are interested in their perspectives about outcome assessment after joint replacement, and how the information collected might be best implemented within their everyday clinical practice.

- Page 8, line 49: can the authors clarify what is meant by 'comparing for similarities and differences'?

The sentence is presently as follows: *Post-surgery qualitative analysis will take a more directed approach, drawing from the information provided by the PROMs, the information collected from the sensors and comparing for similarities and differences.*

Thank you for your recommendation. We have amended the sentence and added a reference to substantiate the change.

Post-surgery qualitative analysis will take a more directed approach so that findings can be triangulated with information provided by the PROMs and from the sensors. The purposes of triangulation is not necessarily to cross validate data but to capture different dimensions of the same phenomenon.[20]

- Page 10, line 15: will the surgeons have seen the PROMs and sensor data prior to the focus group to enable them to compare the data sources (as per Box 3)?

The study is iterative in the design process with continuous clinic input from the two lead clinicians within the team about the parameters of interest for the sensor data. These clinicians will therefore

have seen some of the sensor data and subsequent data visualisations prior to the main focus group. Considerable time and effort will be spent in preparing the two data sources (PROMs and the output of SPHERE) in a presentation form whereby this will serve as effective stimulus for discussion.

- Page 10, line 38: the authors could leave out the SF-12 measure given they are already collecting EQ-5D data and mindful of respondent burden as participants will also be completing the disease-specific measures

Thank you for the suggestion. Through our pilots and feedback from patients there is presently no indication of respondent burden.

- Page 12, line 12: it's not clear why participants will have the capability of deleting data from the SPHERE system: how will this be tracked and how might it affect sensor data completeness?

The option of deleting data from the SPHERE system was a requirement of approval by the supervising Ethics committee, that is giving the participant the autonomy to control the data and potentially remove non-health related-behaviour or data from non-consenting visitors to the home. When data is deleted is logged to gain an idea of magnitude of the data deleted. In practice although the facility is provided participants seldom use it and therefore unlikely to affect data quality.

We have included a sentence within page 11 of the manuscript to state:

Frequency of deletion of data will be logged for monitoring purposes in order to address the magnitude of deletion of data.

- Page 14, Table 1 – the sleep quality questionnaire is not mentioned in this table

Thank you this has now been amended to include the PSQI

- Page 14, Table 1 – am querying whether the First telephone follow-up at 4-9 days post-hospital discharge is needed? This is a very early post-operative time point when most patients will still have significant pain and may not want to undergo the telephone survey (the possibility of missing data should be considered). I think it would be reasonable to make the 6-week telephone call the first post-operative follow-up

Since the advent of “fast-track” or “day case” joint replacement surgery, it has become increasingly common for patients undergoing joint replacement to receive monitoring telephone calls within the first week of discharge from hospital to monitor their progress and attempt to detect any early complications. In the experience of our clinical team, patients are happy with such telephone calls. It is routine practice in the area in which this study is being conducted for patients to attend for routine wound review (either the hospital or GP practice) at between 10 and 14 days postoperatively. At 1-

week post-operation, primary joint replacement patients are typically mobile with the aid of two walking sticks and performing physiotherapy exercises at home 4 times a day. We agree that the utility of some of the PROMs questionnaires (such as questions that rely on the reporting of symptoms over the last 4 weeks) is compromised at this early phase of follow up. We hope that collecting data at this early stage would allow us to explore whether there is useful data that can be collected at this phase by the sensor technology and how this relates to traditional outcome measures.

- Page 15, line 16: it's not clear why a global measure of patient function and mobility will be developed and how this will be used (will this be used to compare to sensor data?)

Thank you for your query. We acknowledge that 'global' measure may be mis-leading, therefore we have changed the wording to objective 3 and the sentence on page 14 (your page 15, line 16) to state:

- iii) To assess whether data from the SPHERE system provides a comparable measure of patient function and mobility than routinely collected standardised patient measures of PROMs.

The development of a comparable measure of patient function and mobility will be informed by the quantitative and qualitative strands of the project.

- Page 16: while the disruption of sleep data can be compared directly to the sleep quality questionnaire data, I'm concerned that the activity data collected from the sensors is really measuring something different to that collected with the other PROMs tools (I note that no standardised physical activity tool is being used). Activity levels are a very different construct to hip- and knee-related function (collected by the OHS/OKS, with regard to specific functional activities including getting in/out of a car or public transport, doing the household shopping, pain in bed at night). The EQ-5D only includes 1 mobility item and the HOOS/KOOS quality of life subscales don't refer specifically to mobility or physical activity at all. Given this, I'm not sure that it makes for a fair comparison and would like the authors to clarify exactly how they will be comparing the sensor data and provide justification for why this is a reasonable approach

Our objectives are to determine whether the sensor data can help to provide complimentary outcomes of passive monitoring. It is anticipated that through the variety of sensors within the system measuring different elements (temperature, humidity, light, orientation/tilt, appliance monitoring), Our aims of HEmiSPHERE draw on looking for parallels of the sensor data that correlate to the PROMs drawing on specific to items on the OHS/OKS such as those the reviewer mentions e.g. pain in bed at night, getting in and out of a chair), however another large feature of this study is to determine observations within the sensor data that may be very different to the PROMs. Through the latter establishing metrics that provide something beyond what PROMs captures.

Ethics and dissemination

- Page 18, line 20: this should read 'a single site study'

This is on the manuscript, Thank you.

Other points:

- A brief section summarising the study limitations should also be included
- It would be also valuable to report costs data if available (eg average cost per patient, with regard to the costs of equipment, installation and ongoing monitoring and IT support)

Thank you for your points, we now include a study limitations section. The study is an exploratory study to determine amongst many things costs data. This forms part of wider programme within SPHERE and therefore goes beyond the purposes of the present study.

VERSION 2 – REVIEW

REVIEWER	Ilana Ackerman Monash University, Australia
REVIEW RETURNED	02-May-2018
GENERAL COMMENTS	The authors have done a good job addressing my earlier comments and suggestions. The only further minor change I would suggest is to include a sentence somewhere in the manuscript pertaining to costs data (eg average cost per patient, with regard to the costs of equipment, installation and ongoing monitoring and IT support), either in the Methods or Limitations, acknowledging that these data would be valuable and will be collected as part of the broader research program, as readers are likely to be interested in this aspect.

VERSION 2 – AUTHOR RESPONSE

In response to Reviewer 1, we agree cost-effectiveness is a vital part of this work. This is a feasibility study which will inform our future research. We have included a statement within the limitations section (page 16) of the manuscript containing information about cost effectiveness forming an integral part of the wider programme of research for SPHERE.